# A Pseudo-Semantic Loss for Autoregressive Models with Logical Constraints

**Kareem Ahmed**
Department of Computer Science
University of California, Los Angeles
ahmedk@cs.ucla.edu

**Kai-Wei Chang**
Department of Computer Science
University of California, Los Angeles
kwchang@cs.ucla.edu

**Guy Van den Broeck**
Department of Computer Science
University of California, Los Angeles
guyvdb@cs.ucla.edu

## Abstract

Neuro-symbolic AI bridges the gap between purely symbolic and neural approaches to learning. This often requires maximizing the likelihood of a symbolic constraint w.r.t. the neural network's output distribution. Such output distributions are typically assumed to be fully-factorized. This limits the applicability of neuro-symbolic learning to the more expressive autoregressive distributions, e.g., transformers. Under such distributions, computing the likelihood of even simple constraints is #P-hard. Instead of attempting to enforce the constraint on the entire output distribution, we propose to do so on a random, local approximation thereof. More precisely, we optimize the likelihood of the constraint under a pseudolikelihood-based approximation centered around a model sample. Our approximation is factorized, allowing the reuse of solutions to sub-problems—a main tenet for efficiently computing neuro-symbolic losses. Moreover, it is a local, high-fidelity approximation of the likelihood, exhibiting low entropy and KL-divergence around the model sample. We evaluate our approach on Sudoku and shortest-path prediction cast as autoregressive generation, and observe that we greatly improve upon the base model's ability to predict logically-consistent outputs. We also evaluate on the task of detoxifying large language models. Using a simple constraint disallowing a list of toxic words, we are able to steer the model's outputs away from toxic generations, achieving SoTA detoxification compared to previous approaches.

## 1 Introduction

Neuro-symbolic AI aims to consolidate purely statistical approaches, chiefly using neural networks, with purely symbolic approaches for learning and reasoning. It has thus far shown great promise in addressing many of the shortcomings of both paradigms, developing scalable approaches that learn from unstructured data while leveraging domain knowledge to ensure the explainability, trusted behavior as well as reduce the amount of labeled data required by typically data-hungry deep neural networks.

More specifically, a common approach to neuro-symbolic learning consists in injecting knowledge regarding the underlying problem domain into the training process as an auxiliary form of supervision. Such knowledge typically takes the form of a sentence in logic, and relates the outputs of the neural network, delineating assignments to the output variables that constitute a valid object from those that do not. For instance, only an assignment to the cells of a Sudoku puzzle such that each row, column, and $3 \times 3$ square contain all of the digits from $1$ to $9$ constitutes a valid Sudoku solution.

37th Conference on Neural Information Processing Systems (NeurIPS 2023).

Injecting such knowledge into training is typically achieved by maximizing the probability of the constraint—a sum of product of probabilities of the solutions to the constraint—w.r.t. to the network's distribution. There the outputs are assumed to be conditionally independent given the learned features, and therefore the distribution over the solutions of the constraint assumed to be fully-factorized.

In this paper we move beyond fully-factorized output distributions and towards autoregressive ones, including those induced by large language models such as GPT [35], where the output at any given time step depends on the outputs at all previous time steps. Computing the probability of an arbitrary constraint under fully-factorized output distributions is #P-hard. Intuitively, the hardness of the problem can be attributed to the possibly exponentially-many solutions of the constraint. Under an autoregressive distribution, however, computing the probability of even a single literal as a constraint is #P-hard [37]. That is, under autoregressive distributions, the hardness of computing the probability of an arbitrary constraint is now due to two distinct factors: the hardness of the logical constraint as well as the

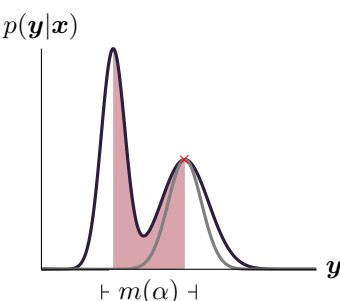

$$p(\boldsymbol{y}|\boldsymbol{x})$$

$$\vdash m(\alpha) \dashv \qquad \boldsymbol{y}$$

Figure 1: **Our approach in a nutshell**. Given a data point $x$, we approximate the likelihood of the constraint $\alpha$ (area shaded in pink) with the pseudolikelihood (shown in gray) of the constraint in the neighborhood of a sample (denoted ×), where $m(\alpha)$ denotes the region of the constraint support.

hardness of the distribution. Throughout this paper, we will assume the inherent hardness of the constraint can be sidestepped: for many applications, we can come up with compact representations of the constraint's solutions that are amenable to computing its probability under the fully-factorized distribution efficiently (cf. Section 3.1). For problems where such compact representations are unavailable, we can fall back to approximate representations of the constraint [4].

Unlike previous works that are only able to approximately handle simple constraints under relaxations of autoregressive distributions [14, 50, 21, 48], our approach injects non-trivial constraints, that don't easily factorize, as part of the training process, computing the probability of the constraint exactly w.r.t. an approximate distribution. Concretely, we approximate the likelihood of the constraint w.r.t. the autoregressive distribution with its probability in a local pseudolikelihood distribution—a product of conditionals—centered around a model sample. This leads to a factorizable objective which allows us to efficiently compute the probability of constraints by reusing solutions to common sub-problems. Experiments show our approximation is low-entropy, allocating most of its mass around the sample, and has low KL- divergence from the true distribution. Intuitively, we want to stay close to the sample to ensure high fidelity, while retaining a distribution to ensure differentiability and maximum generality within tractability bounds. An overview of our proposed approach is depicted in Figure 1.

Empirically, we start by evaluating our approach on the tasks of solving a Sudoku puzzle and generating a shortest path in a given Warcraft map where, conditioned on the input puzzle (map, resp.), the neural network autoregressively generates a Sudoku solution (shortest path, resp.), taking into account generations at previous time steps. We observe that our autoregressive models improve upon the non-autoregressive baselines, and that our approach leads to models whose predictions are even more accurate, and even more likely to satisfy the constraint. Lastly, we evaluated our approach on the challenging task of detoxifying pretrained large language models where the aim is to move the model's distribution away from toxic generations and towards nontoxic ones without sacrificing the model's overall language modeling abilities. We show that, perhaps surprisingly, using only a simple constraint disallowing a list of toxic words, the model exhibits a great reduction in the toxicity of the generated sentences, as measured using the perspective API[1], at almost no cost in terms of the model's language modeling capabilities, measured in perplexity; to our knowledge the first use of logical constraints in such task. Our code is available at github.com/UCLA-StarAI/PseudoSL.

**Contribution** In summary, we propose approximating the likelihood of the constraint w.r.t. the model parameters with the pseudolikelihood of the constraint centered around a model sample. Our approach can be thought of as penalizing the neural network for all the probability mass it allocates to the local perturbations of a model sample that volate the logical constraint. We empirically demonstrate that our approach leads to models that are more consistent with the constraint.

---

[1] https://www.perspectiveapi.com/

## 2 Background

### 2.1 Notation

We write uppercase letters ($X$, $Y$) for Boolean variables and lowercase letters ($x$, $y$) for their instantiation ($Y = 0$ or $Y = 1$). Sets of variables are written in bold uppercase ($\mathbf{X}$, $\mathbf{Y}$), and their joint instantiation in bold lowercase ($\boldsymbol{x}$, $\boldsymbol{y}$). A literal is a variable ($Y$) or its negation ($\neg Y$). A logical sentence ($\alpha$ or $\beta$) is constructed from variables and logical connectives ($\wedge$, $\vee$, etc.), and is also called a (logical) formula or constraint. A state or world $\boldsymbol{y}$ is an instantiation to all variables $\mathbf{Y}$. A state $\boldsymbol{y}$ satisfies a sentence $\alpha$, denoted $\boldsymbol{y} \models \alpha$, if the sentence evaluates to true in that world. A state $\boldsymbol{y}$ that satisfies a sentence $\alpha$ is also said to be a model of $\alpha$. We denote by $m(\alpha)$ the set of all models of $\alpha$. The notation for states $\boldsymbol{y}$ is used to refer to an assignment, the logical sentence enforcing the assignment, or the binary output vector capturing the assignment, as these are all equivalent notions. A sentence $\alpha$ entails another sentence $\beta$, denoted $\alpha \models \beta$, if all worlds that satisfy $\alpha$ also satisfy $\beta$.

### 2.2 A Probability Distribution over Possible Structures

Let $\alpha$ be a logical sentence defined over Boolean variables $\mathbf{Y} = \{Y_{11}, \ldots, Y_{nk}\}$, where $n$ denotes the number of time steps in the sequence, and $k$ denotes the number of possible classes at each step.

The neural network's outputs induce a probability distribution $p(\cdot)$ over possible states $\boldsymbol{y}$. However, the neural network will ensure that, for each time step $i$, there is exactly one class being predicted in each possible state. That is, exactly one Boolean variable $\{Y_{i1}, \ldots, Y_{ik}\}$ can be set to true for each time step $i$. We will use $\boldsymbol{y}_i$ to denote that variable $Y_{ij}$ is set to true in state $\boldsymbol{y}$. More precisely, we let $\boldsymbol{y}_i \in \{0, 1\}^k$ be the one-hot encoding of $Y_{ij}$ being set to 1 among $\{Y_{i1}, \ldots, Y_{ik}\}$. By the chain rule, the probability assigned by the autoregressive neural network to a state $\boldsymbol{y}$ is then defined as

$$p(\boldsymbol{y}) = \prod_{i=1}^{n} p(\boldsymbol{y}_i \mid \boldsymbol{y}_{<i}), \tag{1}$$

where $\boldsymbol{y}_{<i}$ denotes the prefix $\boldsymbol{y}_1, \ldots, \boldsymbol{y}_{i-1}$. The most common approaches [30, 46, 16] to neuro-symbolic learning assume the conditional independence of the network outputs given the learned embeddings. More precisely, let $f$ be a neural network that maps inputs $\boldsymbol{x}$ to $M$-dimensional embeddings $\boldsymbol{z} = f(\boldsymbol{x})$. Under such assumption, we obtain the *fully-factorized* distribution

$$p(\boldsymbol{y} \mid \boldsymbol{z}) = \prod_{i=1}^{n} p(\boldsymbol{y}_i \mid \boldsymbol{z}). \tag{2}$$

We no longer have a notion of ordering under the fully-factorized distribution—and each possible $p(\boldsymbol{y}_i \mid \boldsymbol{z})$ is computed as $\sigma(\mathbf{w}_i^\top \boldsymbol{z})$ where $\mathbf{w}_i \in \mathbb{R}^M$ is a vector of parameters and $\sigma(x)$ is the softmax function. The appeal of such distribution is that it enables the tractability of many reasoning tasks, but the downside is that it dismisses any correlation between the output labels. As we will show in our experimental section (cf. Section 5), using autoregressive distributions, even simple ones such as LSTMs, already outperforms a neural network where the labels are assumed to be independent.

### 2.3 Neuro-Symbolic Losses

In neuro-symbolic learning, we often assume access to symbolic knowledge connecting the different outputs of a neural network, typically in the form of a constraint (or sentence) $\alpha$ in Boolean logic. We are concerned with maximizing the likelihood of the constraint $\alpha$ w.r.t. the network's parameters $\boldsymbol{\theta}$:

$$\operatorname*{argmax}_{\boldsymbol{\theta}} p_{\boldsymbol{\theta}}(\alpha) = \operatorname*{argmax}_{\boldsymbol{\theta}} \mathbb{E}_{\boldsymbol{y} \sim p_{\boldsymbol{\theta}}} \left[ \mathbb{1}\{\boldsymbol{y} \models \alpha\} \right] = \operatorname*{argmax}_{\boldsymbol{\theta}} \sum_{\boldsymbol{y} \models \alpha} p_{\boldsymbol{\theta}}(\boldsymbol{y}), \tag{3}$$

where, with a slight abuse of notation, we omit the inputs $\boldsymbol{x}$. The expectation in Equation (3) quantifies how close the neural network comes to satisfying the constraint. It does so by reducing the problem of probability computation to weighted model counting (WMC): summing up the models of $\alpha$, each weighted by its likelihood under $p$. The negative logarithm of this expectation yields a loss function called semantic loss [46]. Depending on how one chooses to compute said expectation, we recover different approaches. T-norms [28] make various assumptions, for instance, that the clauses of the

constraint are independent [36]. Ahmed et al. [1] estimate the expectation by sampling, using various gradient estimators for learning. Xu et al. [46] compute the objective exactly, leveraging knowledge compilation techniques that exploit the structure embedded in the solution space. They obtain a target representation (a circuit) in which computing the expectation in Equation (3) using dynamic programming is linear in the size of the target representation (the number of circuit edges). We note that computing this expectation is, for arbitrary constraints, #P-hard [39, 40]. Indeed the size of the compiled circuit can grow exponentially in the constraint. In practice, we can obtain compact circuits for many constraints of interest, or effectively decompose the constraints as an approximation [4].

## 3   Pseudo-Semantic Loss

Unfortunately, as previously mentioned, moving beyond the fully-factorized distribution, we are faced with another source of intractability: the hardness of the distribution w.r.t. which the expectation in Equation (3) is being computed. Assuming a deep generative model whose distribution $p$ can capture a Bayesian network distribution, the problem of computing even a single marginal—i.e., the marginal probability of a single variable—is known to be #P-hard [37]. This class of models includes the autoregressive distribution. Intuitively, a constraint might have exponentially-many solutions, yet lend itself nicely to reusing of solutions to sub-problems, and therefore a tractable calculation of the expectation in Equation (3). An example being the $n$ choose $k$ constraint [5], where the expectation in Equation (3) can be computed in quadratic time under the fully-factorized distribution, despite having a normally-prohibitive number of solutions. Moving away from the fully-factorized distribution, however, entails that in the worst case, we would need to compute a sub-problem combinatorial number of times—for all possible sequences—for the exponentially many solutions of the constraint.

To sidestep the intractability of the expectation in Equation (3), as a first step, we consider the *pseudolikelihood* $\tilde{p}(\cdot)$ of a set of parameters given an assignment [6], as a surrogate for its likelihood i.e.,

$$p(\boldsymbol{y}) \approx \tilde{p}(\boldsymbol{y}) := \prod_i p(\boldsymbol{y}_i \mid \boldsymbol{y}_{-i}), \tag{4}$$

where $\boldsymbol{y}_{-i}$ denotes $\boldsymbol{y}_1, \ldots, \boldsymbol{y}_{i-1}, \boldsymbol{y}_{i+1}, \ldots, \boldsymbol{y}_n$. Consequently, we can consider *the pseudolikelihood* of a set of parameters given a logical constraint $\alpha$ as a surrogate for its true likelihood i.e.,

$$p(\alpha) \approx \tilde{p}(\alpha) = \mathbb{E}_{\boldsymbol{y} \sim \tilde{p}}\left[\mathbb{1}\{\boldsymbol{y} \models \alpha\}\right] = \sum_{\boldsymbol{y} \models \alpha} \tilde{p}(\boldsymbol{y}). \tag{5}$$

Intuitively, the pseudolikelihood objective aims to measure our ability to predict the value of each variable given a full observation of all other variables. The pseudolikelihood objective is attempting to match all of the model's conditional distributions to the conditional distributions computed from the data. If it succeeds in matching them exactly, then a Gibbs sampler run on the model's conditional distributions attains the same invariant distribution as a Gibbs sampler run on the true data distribution.

On its own, the above would still not be sufficient to ensure the tractability of the expectation in Equation (3). Intuitively, different solutions depend on different sets of conditionals, meaning we would have to compute the probabilities of many of the solutions of the constraint from scratch.

Instead, *we compute the pseudolikelihood of the constraint in the neighborhood of a model sample*[2]

$$\tilde{p}(\alpha) = \mathbb{E}_{\boldsymbol{y} \sim \tilde{p}}\left[\mathbb{1}\{\boldsymbol{y} \models \alpha\}\right] \approx \mathbb{E}_{\boldsymbol{y} \sim p}\mathbb{E}_{\tilde{\boldsymbol{y}} \sim \tilde{p}_{\boldsymbol{y}}}\left[\mathbb{1}\{\tilde{\boldsymbol{y}} \models \alpha\}\right] = \mathbb{E}_{\boldsymbol{y} \sim p}\,\tilde{p}_{\boldsymbol{y}}(\alpha) = \mathbb{E}_{\boldsymbol{y} \sim p}\sum_{\tilde{\boldsymbol{y}} \models \alpha} \tilde{p}_{\boldsymbol{y}}(\tilde{\boldsymbol{y}}), \tag{6}$$

$$\text{where } \tilde{p}_{\boldsymbol{y}}(\tilde{\boldsymbol{y}}) := \prod_i p(\tilde{\boldsymbol{y}}_i \mid \boldsymbol{y}_{-i}) \tag{7}$$

which can be seen as the pseudolikelihood $\tilde{p}(\cdot)$ of an assignment in the neighborhood of a sample $\boldsymbol{y}$. Crucially this distribution is fully factorized, making it amenable to neuro-symbolic loss functions.

**Definition 3.1** (Pseudo-Semantic Loss). Let $\alpha$ be a sentence in Boolean logic, and let $\tilde{p}_{\boldsymbol{y}}(\cdot)$ be the pseudolikelihood function parameterized by $\boldsymbol{\theta}$ and centered around state $\boldsymbol{y}$, as defined in Equation (7). Then, we define the pseudo-semantic loss between $\alpha$ and $\boldsymbol{\theta}$ to be

$$\mathcal{L}^{\mathsf{SL}}_{\mathsf{pseudo}}(\alpha, p_{\boldsymbol{\theta}}) := -\log \mathbb{E}_{\boldsymbol{y} \sim p}\,\tilde{p}_{\boldsymbol{y}}(\alpha) = -\log \mathbb{E}_{\boldsymbol{y} \sim p}\sum_{\tilde{\boldsymbol{y}} \models \alpha} \tilde{p}_{\boldsymbol{y}}(\tilde{\boldsymbol{y}}). \tag{8}$$

---

[2]We sample $y_1$ conditioned on the beginning-of-sentence token, then $y_2$ conditioned on the sampled $y_1$, followed by $y_3$ conditioned on both $y_1$ and $y_2$ and so on until we the end-of-sentence token is sampled.

Intuitively, our pseudo-semantic loss between $\alpha$ and $p_{\boldsymbol{\theta}}$ can be thought of as penalizing the neural network for all the local perturbations $\tilde{\boldsymbol{y}}$ of the model sample $\boldsymbol{y}$ that violate the logical constraint $\alpha$.

## 3.1 Tractable Expectation Computations

We appeal to knowledge compilation techniques—a class of methods that transform, or *compile*, a logical theory into a *tractable circuit* target form, which represents functions as parameterized computational graphs. By imposing certain structural properties on these computational graphs, we enable the tractable computation of certain classes of probabilistic queries over the encoded functions. As such, circuits provide a language for building and reasoning about tractable representations.

**Logical Circuits** More formally, a *logical circuit* is a directed, acyclic computational graph representing a logical formula. Each node $n$ in the DAG encodes a logical sub-formula, denoted $[n]$. Each inner node in the graph is either an AND or an OR gate, and each leaf node encodes a Boolean literal ($Y$ or $\neg Y$). We denote by $\mathsf{in}(n)$ the set of $n$'s children, that is, the operands of its logical gate.

**Structural Properties** As already alluded to, circuits enable the tractable computation of certain classes of queries over encoded functions granted that a set of structural properties are enforced.

A circuit is *decomposable* if the inputs of every AND gate depend on disjoint sets of variables i.e. for $\alpha = \beta \wedge \gamma$, $\mathsf{vars}(\beta) \cap \mathsf{vars}(\gamma) = \varnothing$. Intuitively, decomposable AND nodes encode local factorizations over variables of the function. For simplicity, we assume that decomposable AND gates always have two inputs, a condition enforceable on any circuit in exchange for a polynomial size increase [41, 32].

A second useful property is *smoothness*. A circuit is *smooth* if the children of every OR gate depend on the same set of variables i.e. for $\alpha = \bigvee_i \beta_i$, we have that $\mathsf{vars}(\beta_i) = \mathsf{vars}(\beta_j) \, \forall i, j$. Decomposability and smoothness are a sufficient and necessary condition for tractable integration over arbitrary sets of variables in a single pass, as they allow larger integrals to decompose into smaller ones [8].

Furthermore, a circuit is said to be *deterministic* if, for any input, at most one child of every OR node has a non-zero output i.e. for $\alpha = \bigvee_i \beta_i$, we have that $\beta_i \wedge \beta_j = \bot$ for all $i \neq j$. Similar to decomposability, determinism induces a recursive partitioning of the function, but over the support, i.e. satisfying assignments, of the function, rather than the variables. Determinism, taken together with smoothness and decomposability, allows us to tractably compute a constraint probability [10]. Given a smooth, deterministic and decomposable logical circuit $c_\alpha$ encoding a constraint $\alpha$[3] we can compute the probability $p(\alpha)$ w.r.t. a distribution $p$ that factorizes by feeding the probability of each literal at the corresponding leaf node and evaluating the circuit upwards, taking sums at OR nodes and products at AND nodes. Figure 2 shows an example of computing the probability of such a circuit.

## 3.2 The Algorithm

We will now give a walk through of computing our pseudo-semantic loss. We note that our algorithm is implemented in log-space to preserve numerical stability and uses PyTorch [31]. Our full algorithm is shown in Algorithm 1. We sample an assignment $\boldsymbol{y} \sim p_{\boldsymbol{\theta}}$ from the model (line 4). We now need to compute the pseudolikelihood of the sample

$$\log \tilde{p}_\theta(\boldsymbol{y}) = \sum_i \log p(\boldsymbol{y}_i \mid \boldsymbol{y}_{-i})$$
$$= \sum_i \log p(\boldsymbol{y}_i, \boldsymbol{y}_{-i}) - \mathop{\mathrm{LSE}}_{\boldsymbol{y}_i'} \log p(\boldsymbol{y}_i', \boldsymbol{y}_{-i}),$$

where LSE is the logsumexp function. That is, for every element in the sequence, we need to marginalize over all categories $\boldsymbol{y}_i'$. This entails, for every element in the sampled sequence, we need to substitute each of the categories (lines

---

**Algorithm 1** $\mathcal{L}^{\mathsf{SL}}_{\mathsf{pseudo}}(\alpha; p_{\boldsymbol{\theta}})$

1: **Input**: Logical constraint $\alpha$ and model $p_{\boldsymbol{\theta}}$.
2: **Output**: Pseudo-semantic loss of $\alpha$ w.r.t. $\boldsymbol{\theta}$
3: // Obtain sample $y$ from $p_{\boldsymbol{\theta}}$
4: $\boldsymbol{y} \sim p_{\boldsymbol{\theta}}$
5: // Get sequence length and num. of categories
6: seq, cats = $\boldsymbol{y}$.shape()
7: // Expand the batch to contain all perturbations
8: // of $\boldsymbol{y}$ that are a Hamming distance of 1 away
9: $\boldsymbol{y} = \boldsymbol{y}$.expand(seq, cats)
10: $\boldsymbol{y}[:, \mathrm{range}(\mathrm{seq}), :, \mathrm{range}(\mathrm{seq})] = \mathrm{range}(\mathrm{cats})$
11: // Evaluate expanded samples through model
12: $\log p_{\boldsymbol{\theta}} = p_{\boldsymbol{\theta}}(\boldsymbol{y}).\mathrm{log\_softmax}(\mathrm{dim} = -1)$
13: // Compute the conditional probabilities:
14: // $\log \tilde{p}_{\boldsymbol{\theta}}[i][j] = \log p_{\boldsymbol{\theta}}(\boldsymbol{y}_j | \boldsymbol{y}_{-j})$
15: $\log \tilde{p}_{\boldsymbol{\theta}} = \log p_{\boldsymbol{\theta}} - \log p_{\boldsymbol{\theta}}.\mathrm{logsumexp}(\mathrm{dim}=-1)$
16: // Compute the probability of $\alpha$ under $\tilde{p}_{\boldsymbol{y}}$
17: // by propagating the conditionals through $c_\alpha$
18: **return** $-\log \tilde{p}_{\boldsymbol{y}}(\alpha)$

---

[3] Such a circuit can always be constructed, (Appendix A) although it can grow exponentially in the worst case.

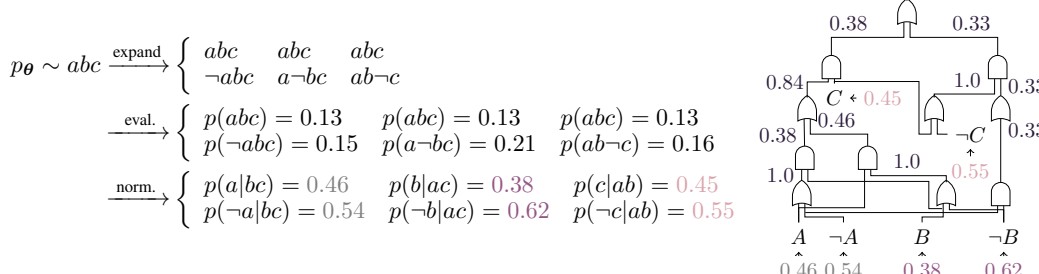

$$p_{\boldsymbol{\theta}} \sim abc \xrightarrow{\text{expand}} \begin{cases} abc & abc & abc \\ \neg abc & a\neg bc & ab\neg c \end{cases}$$

$$\xrightarrow{\text{eval.}} \begin{cases} p(abc) = 0.13 & p(abc) = 0.13 & p(abc) = 0.13 \\ p(\neg abc) = 0.15 & p(a\neg bc) = 0.21 & p(ab\neg c) = 0.16 \end{cases}$$

$$\xrightarrow{\text{norm.}} \begin{cases} p(a|bc) = 0.46 & p(b|ac) = 0.38 & p(c|ab) = 0.45 \\ p(\neg a|bc) = 0.54 & p(\neg b|ac) = 0.62 & p(\neg c|ab) = 0.55 \end{cases}$$

Figure 2: **An example of our pipeline.** (Left) We start by sampling an assignment from the model $p_{\boldsymbol{\theta}}$. Our goal is to compute the pseudolikelihood of the model sample—the product of the sample's conditionals. We start by expanding the model sample to include all samples that are a Hamming distance of 1 away from the sample. We proceed by (batch) evaluating the samples through the model, obtaining the joint probability of each sample. We then normalize along each column, obtaining the conditionals. (Right) A logical circuit encoding constraint $(\text{Cat} \implies \text{Animal}) \land (\text{Dog} \implies \text{Animal})$, with variable $A$ mapping to Cat, variable $B$ mapping to dog and variable $C$ mapping to Animal. To compute the pseudolikelihood of the constraint in the neighborhood of the sample $abc$, we feed the computed conditional at the corresponding literals. We push the probabilities upwards, taking products at AND nodes and sums at OR nodes. The number accumulated at the root of the circuit is the pseudolikelihood of the constraint in the neighborhood of the sample $abc$.

9-10) and compute the probability of the sample under the model (line 12), obtaining sequence length $\times$ number of categories sequences. Now we can compute the log-conditional probabilities $\log p(\boldsymbol{y}_i \mid \boldsymbol{y}_{-i})$. We marginalize over the categories $\boldsymbol{y}'_i$ to obtain the log-marginal $\log p(\boldsymbol{y}_{-i}) = \text{LSE}\,\boldsymbol{y}'_i(\log p(\boldsymbol{y}'_i, \boldsymbol{y}_{-i}))$. We then condition the probability of every sequence by subtracting the log-marginals i.e., $\log p(\boldsymbol{y}_i, \boldsymbol{y}_{-i}) - \log p(\boldsymbol{y}_{-i})$ (line 15). We use these conditionals to compute the pseudolikelihood assigned by the neural network to local perturbations of the model sample $y$ that satisfy the constraint (line 18). As per Section 3.1, we can compute the pseudolikelihood of a constraint $\alpha$ locally around the sample $y$ by pushing the computed conditionals at the respective input nodes of $c_{\alpha}$, propagating them through the circuit, taking sums and products, and reading the value at the circuit root. Figure 2 shows a toy example run of our algorithm in non-log space.

## 4 Related Work

In an acknowledgment to the need for both symbolic as well as sub-symbolic reasoning, there has been a plethora of recent works studying how to best combine neural networks and logical reasoning, dubbed *neuro-symbolic AI*. The focus of such approaches is typically making probabilistic reasoning tractable through first-order approximations, and differentiable, through reducing logical formulas into arithmetic objectives, replacing logical operators with their fuzzy t-norms, and implications with inequalities [22, 36, 13].

Another class of neuro-symbolic approaches have their roots in logic programming. Deep-ProbLog [27] extends ProbLog, a probabilistic logic programming language, with the capacity to process neural predicates, whereby the network's outputs are construed as the probabilities of the corresponding predicates. This simple idea retains all essential components of ProbLog: the semantics, inference mechanism, and the implementation. In a similar vein, Dai et al. [9] combine domain knowledge specified as purely logical Prolog rules with the output of neural networks, dealing with the network's uncertainty through revising the hypothesis by iteratively replacing the output of the neural network with anonymous variables until a consistent hypothesis can be formed. Bošnjak et al. [7] present a framework combining prior procedural knowledge, as a Forth program, with neural functions learned through data. The resulting neural programs are consistent with specified prior knowledge and optimized with respect to data.

Diligenti et al. [11] and Donadello et al. [12] use first-order logic to specify constraints on outputs of a neural network. They employ fuzzy logic to reduce logical formulas into differential, arithmetic objectives denoting the extent to which neural network outputs violate the constraints, thereby supporting end-to-end learning under constraints. Xu et al. [46] introduced semantic loss, which circumvents the shortcomings of fuzzy approaches, while still supporting end-to-end learning under

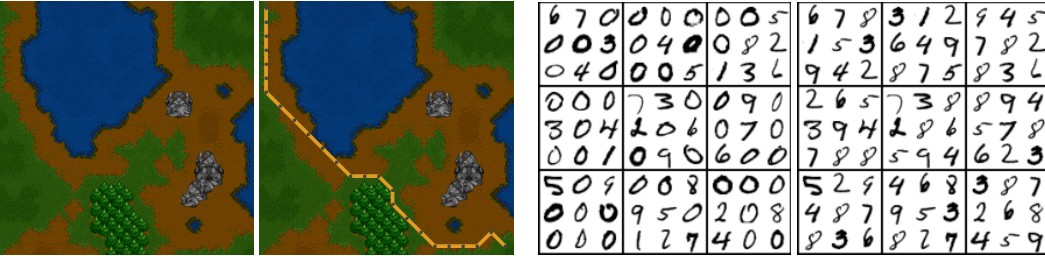

Figure 3: **Example inputs and groundtruth labels for two of the three tasks considered in our experimental evaluation.** (Left) Example Warcraft terrain map and a possible (non-unique) minimum-cost shortest path. (Right) Example Sudoku puzzle and its corresponding (unique) solution.

constraints. More precisely, *fuzzy reasoning* is replaced with *exact probabilistic reasoning*, made possible by compiling logical formulae into structures supporting efficient probabilistic queries.

There has recently been a plethora of approaches ensuring consistency by embedding the constraints as predictive layers, including semantic probabilistic layers (SPLs) [2], MultiplexNet [20] and C-HMCNN [16]. Much like semantic loss [46], SPLs maintain sound probabilistic semantics, and while displaying impressive scalability to real world problems, but might struggle with encoding harder constraints. MultiplexNet is able to encode only constraints in disjunctive normal form, which is problematic for generality and efficiency as neuro-symbolic tasks often involve an intractably large number of clauses. HMCCN encodes label dependencies as fuzzy relaxation and is the current state-of-the-art model for hierarchical multi-label classification [16], but, similar to its recent extension [17], is restricted to a certain family of constraints.

A related line of research focuses on constrained text generation, modifying the decoding algorithm to inject *lexical* constraints into the beam search process. Such methods include constrained beam search [34], NeuroLogic Decoding [25] and A*esque NeuroLogic Decoding [26]. And although they can be easily applied to various language models without training, these search-based methods can be inefficient as they suffer from large search spaces. Recent works like NADO [29] and FUDGE [47] train auxiliary neural models to provide token-level guidance for autoregressive generation. In a similar vein, GeLaTo [49] augments a large language model with guidance from a tractable probabilistic model to guarantee the keyword tokens are part of the generated sentence while retaining its fluency. Another family of approaches that enforce keyword-type constraints are insertion-based language models [24, 38], where the initial sequences only consist of the desired keywords and the transition phrases are repeatedly inserted to complete the sentences.

Throughout this paper, we assumed that the constructing a logical circuit from a logical formula was easy. This is, in general, not the case. Ahmed et al. [4] offer an approach, by assuming the sub-problems are independent, and iteratively relaxing the independence assumption according to the sub-problems that most violate that assumption as measured using the conditional mutual information.

## 5    Experimental Evaluation

We evaluate our pseudo-semantic loss on several tasks, spanning a number of domains. We start by evaluating on Warcraft shortest-path finding, where we are given an image of a Warcraft tilemap, and are tasked with *autoregressively* generating one of the potentially many minimum-cost paths between two end points conditioned on the map, where the cost is determined by the *underlying* cost of the tiles spanned by the path. We move on to evaluating on the classic, yet challenging, task of solving a $9 \times 9$ Sudoku puzzle where, once again, the generation proceeds autoregressively, conditioned on the input Sudoku puzzle. It is worth noting that such tasks have been considered as a test bed for other neuro-symbolic approaches before, but never from an autoregressive generation perspective.

We also evaluate on the task of large language models (LLMs) detoxification. In this task, we are interested in the generations produced by an LLM when presented by a prompt input by the user. More specifically, we are interested not only in how good these models are at the modeling aspect, but also how *toxic* their outputs might be, a measure which includes sexual explicitness, identity attacks, and profanity, among others. Our goal in this task is then to shift the model's distribution

Table 1: Our experimental results on Sudoku.

| Test accuracy % | Exact | Consistent |
|---|---|---|
| ConvNet | 16.80 | 16.80 |
| ConvNet + SL | 22.10 | 22.10 |
| RNN | 22.40 | 22.40 |
| RNN + PSEUDOSL | **28.20** | **28.20** |

Table 2: Our experimental results on Warcraft.

| Test accuracy % | Exact | Consistent |
|---|---|---|
| ResNet-18 | 55.00 | 56.90 |
| ResNet-18 + SL | 59.40 | 61.20 |
| CNN-LSTM | 62.00 | 76.60 |
| CNN-LSTM + PSEUDOSL | **66.00** | **79.00** |

away from toxic generations, and toward nontoxic ones, all while maintaining its original ability to model text. We believe this to be a timely and important problem due to their recent prevalence and widespread usage coupled with the fact that previous work [15] has found non-negligible amounts of toxic, harmful, and abusive text in the corpora used to train LLMs.

Lastly, we evaluated our approximation's fidelity by comparing the entropy of our local approximation against that of the GPT-2 distribution, as well as how close our approximation is to the true likelihood in the proximity of the sampled data point as measured by the KL-divergence between the two. All experimental details, hardware specifications, as well as training details are provided in the appendix.

**Warcraft Shortest Path**    For this task, we follow the experimental setting set forth by [33], where our training set consists of $10,000$ terrain maps curated using Warcraft II tileset. Each map encodes a $12 \times 12$ grid superimposed on a Warcraft terrain map, where each vertex is weighted according to the cost of the tile, which in turn depends on type of terrain it represents e.g., earth has lower cost than water. These costs are *not* presented to the network. The task is then to generate a minimum-cost path from the upper left to the lower right vertices, where the cost of a path is defined as the sum of costs of the vertices visted by the edges along the path, and the minimum-cost path is not unique, i.e., there exists many paths with the minimum cost, and are all considered correct. The minimum cost path between the top left and bottom right vertices is encoded as an indicator matrix, and serves as a label. Figure 3 shows an example input to the network, and the input annotated with a possible path.

We use a CNN-LSTM model, where, presented with an image of a terrain map, we use a ResNet18 [19] to obtain a 128 image embedding, which is then passed on to an LSTM with a single layer, a hidden dim of size 512, and at every time step predicts the next edge in the path conditioned on the image embedding and previous edges. The constraint being maximized by pseudo-semantic loss in this task is that the predicted edges form a valid path.

As has been established in previous work [46, 3, 2], the accuracy of predicting individual labels is often a poor indicator of the performance of the neural network in neuro-symbolic settings, where we are rather more interested in the accuracy of our predicted structure object *exactly* matching the groundtruth label , e.g., *is the prediction a shortest path?*, a metric which we denote "Exact" in our experiments, as well as the accuracy of predicting objects that are *consistent* with the constraint, e.g., *is the prediction a valid path?*, a metric denoted "Consistent". Our results are shown in Table 2.

As alluded to repeatedly throughout the course of the paper, the first observation is that using an autoregressive model to predict the shortest path in the grid, even a simple single layer LSTM outperforms both a ResNet-18, as well a ResNet-18 trained with semantic loss, improving the exact match from $55.00\%$ and $59.40\%$ to $62.00\%$, and greatly improving the consistency of the predicted paths to $76.00\%$, an improvement by almost $15\%$. We also see that using our pseudo-semantic loss, denoted PSEUDOSL, we improve the exact and consistent accuracies to $66.00\%$ and $79.00\%$, resp.

**Sudoku**    Next, we consider the task of predicting a solution to a given Sudoku puzzle. Here the task is, given a $9 \times 9$ partially-filled grid of numbers to fill in the remaining cells such that the entries each row, column, and $3 \times 3$ square are unique i.e., each number from 1 to 9 appears exactly once. We use the dataset provided by Wang et al. [43], consisting of 10K Sudoku puzzles, split into 9K training examples, and 1K test samples, all puzzles having 10 missing entries. As our baseline, we use a 5-layer RNN with a hidden dimension of 128, `tanh` non-linearity and a dropout of $0.2$. At each time step, the RNN predicts the next cell given as input a one-hot encoding of the previous cell, and conditioned on the partially filled Sudoku. The constraint being maximized by pseudo-semantic loss is that entries in each row, column, and $3 \times 3$ squares are unique. Our results are shown in Table 1.

Table 3: Evaluation of LLM toxicity and quality across different detoxification methods on GPT-2 with 124 million parameters. Model toxicity is evaluated on the REALTOXICITYPROMPTS benchmark through Perspective API. **Full**, **Toxic** and **Nontoxic** refer to the full, toxic and nontoxic subsets of the prompts, respectively. **PPL** refers to the model perplexity on the WebText validation set. **PPL** of word banning is evaluated on the 50% nontoxic portion of the WebText validation set. In line with previous work [15, 42], we characterize toxicity using two metrics: the **Expected Maximum Toxicity** over 25 generations, and the **Toxicity Probability** of a completion at least once over 25 generations. Setting the probabilities of toxic words to zero sending the perplexity of to infinity. We, therefore, report the perplexity on the 50% least toxic prompts dataset for **Word Banning** variants.

| Models | | Exp. Max. Toxicity ($\downarrow$) | | | Toxicity Prob. ($\downarrow$) | | | PPL ($\downarrow$) |
|---|---|---|---|---|---|---|---|---|
| | | Full | Toxic | Nontoxic | Full | Toxic | Nontoxic | |
| | GPT-2 | 0.44 | 0.62 | 0.39 | 34.11% | 67.27% | 24.85% | 25.85 |
| **Domain-Adaptive** | SGEAT [42] | 0.32 | 0.46 | 0.28 | 14.05% | 35.72% | 7.99% | 28.72 |
| | PseudoSL *(ours)* | **0.29** | 0.38 | **0.27** | 9.80% | 20.07% | 6.93% | 28.14 |
| **Word Banning** | GPT-2 | 0.40 | 0.55 | 0.36 | 27.92% | 57.86% | 19.56% | 22.24 |
| | SGEAT [42] | 0.30 | 0.41 | **0.27** | 10.73% | 27.05% | **6.17%** | 24.91 |
| | PseudoSL *(ours)* | **0.29** | **0.37** | **0.27** | **9.20%** | **18.71%** | 6.55% | 24.19 |

In line with our previous experiment, we observe that, once again, a simple RNN outperforms the non-autoregressive model, as well as the same model augmented with semantic loss, although the difference is not that big with regards to semantic loss. Augmenting that same autoregressive model with pseudo-semantic loss, however, increases the gap to a convolutional network, and the same convolutional network augmented with semantic loss to 11.40 and 7.10, respectively.

**LLM detoxification** Lastly, we consider the task of LLM detoxification. That is, we investigate the effectiveness of logical constraints, enforced using pseudo-semantic loss, at steering the model away from toxic prompted-generations. We choose a *very* simple constraint to be *minimized* by pseudo-semantic loss throughout this task, namely we minimize the probability that any of a list of profanity, slurs, and swear words[4] appear as part of the model generations. Following previous work [15, 42], we evaluate on the REALTOXICITYPROMPTS, a dataset of almost 100k prompts ranging from nontoxic, assigned a toxicity score of 0, to very toxic, assigned a toxicity score of 1. We focus on GPT-2 [35] as a base model for detoxification. As is customary, [15, 42], we use Perspective API, an online automated model for toxic language and hate speech detection, to score the toxicity of our predictions. It returns scores in the range 0 to 1.0, corresponding to nontoxic on the one end, and extremely toxic on the other. Though not without limitations, studies [42, 44] have shown that the toxicity scores from Perspective API are strongly correlated with human evaluations.

We compare GPT-2 against SGEAT [42]—which finetunes GPT-2 on the nontoxic portion of its self generation, performing unconditional text generation and retaining only generations with toxicity $< 0.5$—and against SGEAT augmented with pseudo-semantic loss. We report the *Expected Maximum Toxicity* and the *Toxicity Probability*. The *Expected Maximum Toxicity* measures the worst-case toxicity by calculating the maximum toxicity over 25 generations under the same prompt with different random seeds, and averaging the maximum toxicity over all prompts. *Toxicity Probability* estimates the empirical probability of generating toxic language by evaluating the fraction of times a toxic continuation is generated at least once over 25 generations with different random seeds for all prompts. To understand the impact of detoxification, we evaluate the quality of the LLM using perplexity on the validation split of WebText, used to train GPT-2. Our results are shown in Table 3.

**Domain-Adaptive Training** It was previously shown that SGEAT lowers the toxicity of the generations produced by GPT-2, albeit at a slight cost in terms of perplexity. This is confirmed by our numbers, where we see that SGEAT reduces the average worst-case toxicity as well as the probability of producing a toxic generation when prompted with either toxic or nontoxic prompts. We also observe that using PseudoSL loss alongside SGEAT *further* reduces the overall average worst-case toxicity as well as the probability of producing a toxic generation, while producing a better language model compared to SGEAT. Much of this reduction in toxicity appears to stem primarily from a reduction in the average worst-case toxicity as well as toxicity probability given *toxic prompts*.

---

[4]List downloaded from here.

**Decoding-Time Methods**  We also compared GPT-2, SGEAT and PseudoSL with variants thereof obtained through augmentation with a decoding-time algorithm, *Word Banning* [15]. *Word Banning* sets the probability of generating any of the words from the aforementioned list of profanity, slurs and swearwords to zero during decoding. We also attempted to compare against NeuroLogic decoding, a search-based decoding algorithm utilizing look-ahead heuristics to optimize for not only the probability of the generated sentence but also the lexical constraints being satisfied. However, attempting to run NeuroLogic decoding on the entire dataset of prompts (100k) using the maximum batch size we could fit on a 48GB GPU yielded an estimated time of 165 hours. Considering a randomly-sampled subset of the prompts, we obtained empty generations for at least 30% of the prompts. The *PPL* of word banning goes to infinity as the probabilities of some banned words are set to zero. We, therefore, report the perplexity on the 50% least toxic portion of the prompts dataset. We observe that augmenting all of the domain-adaptive training baselines with Word Banning reduces their average worst-case toxicity as well as the their toxicity probability. SGEAT augmented with Word Banning exhibits lower toxicity, both average worst-case toxicity and toxicity probability, than all other non-PseudoSL variants. Interestingly, even augmented with Word Banning, SGEAT exhibits higher toxicity than the base PseudoSL model. PSeudoSL augmented with Word Banning exhibits the lowest overall toxicity, both average worst-case toxicity and toxicity probability, with the gap to the second best in terms of toxicity probability being particularly stark on the toxic prompts. We also note that, in our evaluation of PPL for Word Banning, having discarded the toxic sentences, assigned a lower probability under our model, the perplexity of our model is now closer to that of GPT-2.

**Fidelity evaluation**  Lastly, we evaluated the fidelity of our approximation. We start comparing the entropy of our approximate distribution to the true distribution. We want this quantity to be low, as it would mean our approximation only considers assignments centered around the model sample. We also evaluate the KL-divergence of our approximate distribution from the true distribution in the neighborhood of a model sample. We want this quantity to be low as well, as it corresponds to how faithful our approximation is to the true distribution in the neighborhood of the model sample. Intuitively, the KL-divergence measures how many extra bits are needed to encode samples from our approximation using a code optimized for GPT-2, and is zero when the two distributions coincide. We find the entropy of GPT-2 is 80.89 bits while the entropy of our approximation is, on average, 35.08 bits. We also find the KL-divergence $D_{\mathrm{KL}}(\tilde{p}_{\boldsymbol{y}} \mid\mid p_{\boldsymbol{\theta}})$ is on average 4.8 bits. That is we only need 4 extra bits, on average, to encode the true distribution w.r.t. our approximation distribution. Intuitively, we want to stay close to the sample to ensure high fidelity, while retaining a distribution to ensure differentiability and maximum generality within tractability bounds.

## 6   Conclusion

In conclusion, we proposed pseudo-semantic loss, a neuro-symbolic loss for learning with logical constraints in deep generative models. Instead of attempting to enforce the constraint on the entire distribution, our approach does so on a local distribution centered around a model sample. One intuition to our approach is that it is analogous to a first-order approximation of a discrete functions with no analytical form. As such, our approach makes use of first-order information to approximate the value of the discrete function about a given point, in our case the model sample. Our approach factorizes, allowing us to efficiently compute the probability of the constraint under such an approximate distribution using the language of tractable logical circuits. Our approach is able to greatly improve the *accuracy* and *consistency* of the baselines on structured-output prediction tasks, and is more effective at reducing the toxicity of GPT-2 compared to SoTA adaptive training approaches.

## Acknowledgments

We would like to thank Honghua Zhang for helpful discussions throughout the course of the project, and for providing the initial GPT sampling code. We also thank Tao Meng for the discussions that inspired the LLM detoxification experiment. The authors would like to thank the reviewers for their insightful feedback towards improving the paper. This work was funded in part by the DARPA Perceptually-enabled Task Guidance (PTG) Program under contract number HR00112220005, DARPA Assured Neuro Symbolic Learning and Reasoning (ANSR) Program under contract number FA8750-23-2-0004, ONR grant N00014-23-1-2780, NSF grants #IIS-1943641, #IIS-1956441, #CCF-1837129, and a gift from RelationalAI. GVdB discloses a financial interest in RelationalAI.

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

# A  Circuit Construction

Any logical formula can be compiled into a smooth, deterministic and decomposable logical circuit: every disjunction factorizes the solution space into mutually exclusive events whereas every conjunction factorizes the function into two sub-functions over disjoint sets of variables. Here is a simple albeit potentially sub-optimal recipe: order variables lexicographically. Alternate OR and AND nodes. An OR node branches on the current variable being true or false, and has two children: a left (right) AND node whose children are the positive (negative) literal and the subtree corresponding to substituting the positive (negative) literal into the formula. Repeat while variables remain. We use the PySDD compiler which outputs circuits satisfying the above properties, in addition to structured-decomposability, which asserts that functions, or constraints, over the same variables decompose in the same manner. We say the above recipe is potentially sub-optimal as we use a fixed variable order. In general, there can be an exponential gap in the size of the logical circuit obtained using the worst and best variable order. Finding the best such order is, in general, NP-hard. However, in practice, compilers (PySDD included) use search heuristics that yield demonstrably-good orders.

# B  Language Detoxification

The experiments were run on a server with an AMD EPYC 7313P 16-Core Processor @ 3.7GHz, 2 NVIDIA RTX A6000, and 252 GB RAM. Our LLM detoxification experiments utilized both GPUs using the Huggingface Accelerate [18] library.

In order to construct our constraint, we start with the list of bad words[5] and their space-prefixed variants[6]. We then tokenize this list of augment bad words, yielding $871$ unique possibly-bad tokens (some tokens are only bad when considered in context with other tokens), in addition to an extra catch-all good token to which remaining tokens map to. Our constraint then disallows all sentences containing any of the words on the augmented list, starting at any of the sentence locations $0$ through len(sentence) - len(word). The code to process the list of words, the code to create the constraint as well as the constraint itself will be released as part of our code.

Similar to SGEAT [42], the SoTA domain-adaptive training approach to detoxification, we finetune our model on self-generations as opposed to any external dataset. More specifically, we unpromptedly generate 100k samples using GPT-2 through Hugging Face [45], which are then filtered through Perspective API, keeping only the $50\%$ most nontoxic portion of the generations. We leverage the curated nontoxic corpus to further fine-tune the pre-trained LLM with standard log-likelihood loss and adapt it to the nontoxic data domain. Unlike the two other tasks where we use model samples, we use the toxic portion of the corpus to which we apply our newly proposed pseudo-semantic loss. The intuition here is that the local perturbations of a toxic sentence are also toxic, and these are exactly the assignments whose probability we would like to penalize.

Our training script is adapted from that provided by Hugging Face[7]. We use a batch size of 16, a learning rate of 1e-5 with the AdamW optimizer [23] with otherwise default parameters. We did a grid search over the pseudo-semantic loss weight in the values $\{0.001, 0.005, 0.01, 0.05, 0.1, 0.5, 1, 2, 4, 8\}$. All other hyperparameters were left unchanged. Similar to [42], we use use nucleus sampling with $p = 0.9$ and a temperature of 1 during generation. A randomized 10k portion of the RealToxicityPrompts dataset was used to determine early stopping.

For only this task, our implementation of the pseudo-semantic loss makes use of top-$k$ to construct the pseudo-likelihood distribution (lines 7-12 in Algorithm 1) due to the lack of computational resources. We constructed our distribution using only the top-10 good words and the top-470 toxic words.

# C  Sudoku

The experiments were run on a server with an AMD EPYC 7313P 16-Core Processor @ 3.7GHz, 2 NVIDIA RTX A6000, and 252 GB RAM. Training utilized only one of the two GPUs.

---

[5]List downloaded from here.

[6]A word will be encoded differently whether it is space-prefixed or not.

[7]Downloaded from here.

We follow the experimental setting and dataset provided by Wang et al. [43], consisting of 10K Sudoku puzzles, split into 9K training examples, and 1K test samples, all puzzles having 10 missing entries. Our model consists of an RNN with an input size of 9, a hidden dimension of 128, 5 layers, a tanh nonlinearity and a dropout of 0.2. We used Adam with default PyTorch parameters and a learning rate of 3e-4. We did a grid search over the pseudo-semantic loss weight in the values $\{0.01, 0.05\}$. Our constraint disallows any solution in which the rows, columns and square are not unique.

## D  Warcraft Shortest Path

The experiments were run on a server with an AMD EPYC 7313P 16-Core Processor @ 3.7GHz, 2 NVIDIA RTX A6000, and 252 GB RAM. Training utilized only one of the two GPUs. We follow the experimental setting and dataset provided by [33]. Our training set consists of $10,000$ terrain maps curated using Warcraft II tileset. We use a CNN-LSTM model for this task. Precisely, a ResNet-18 encodes the map to an embedding of dimension 128. An LSTM with 1 layer, and a hidden size of $512$ then predicts the next edge in the shortest path conditioned on the input map and all previous edges. We used Adam with the default PyTorch parameters and a learning rate of 5e-4. We did a grid search over the pseudo-semantic loss weight in the values $\{0.001, 0.005, 0.01, 0.05, 0.1, 0.5, 1\}$. Our constraint disallows any prediction not a valid path connecting the upper left and lower right vertices.

## E  Broader Impact

The work presented in this paper, pseudo-semantic loss, has a significant potential for positive societal impact. Neuro-symbolic learning moves us closer to models whose behavior is trustworthy, explainable and fair. This extends to critical domains such as autonomous driving, medical diagnosis and financial planning to name a few. Large language models have recently seen an exponential increase in popularity, crossing the threshold of being mere research tools into products that are utilized by the general public. Unfortunately, the same expressivity that renders these models so powerful also puts them outside the reach of current neuro-symbolic approaches. Our proposed approach, pseudo-semantic loss, tackles exactly this problem, and does so efficiently. Namely, it brings neuro-symbolic learning, and the promise of trustworthy, explainable and fair models to LLMs. And we have shown the merits of our approach when applied to LLM detoxification. We must, however, also be cognizant of the potential negative societal impacts. More precisely, in very much the same way that our approach can be used to steer the model away from toxic, or generally inconsistent, outputs it can also be used to steer the model towards toxic and harmful generations.

## F  Limitations

Our approach assumes access to hard symbolic knowledge. Such knowledge is not always available, and is not always easy to capture and express symbolically. Our approach also currently only supports hard symbolic knowledge, whereas often times we might be interested in distributional soft constraints that only hold in expectation. Our approach, while tractable, requires a sufficient amount of memory in order to construct the local distribution centered around the model sample. Lastly, our approach approximates the distribution of the model locally, and although we have empirically shown it's effectiveness on three different tasks, it's not clear what guarantees one can derive in general. We view addressing all of the above limitations as very interesting and impactful future endeavors.

## G   Example Generations

Warning! The following contains explicit and/or triggering content.

Example prompted generations of all the approaches considered can be found at `https://github.com/UCLA-StarAI/PseudoSL/blob/main/generations.pdf`

