# OpenReview forum: "A Pseudo-Semantic Loss for Autoregressive Models with Logical Constraints"
_NeurIPS.cc/2023/Conference — NeurIPS 2023 poster_

### Official Review · Reviewer_P5zX · 2023-06-26

**Soundness:** 3 good
**Presentation:** 3 good
**Contribution:** 3 good
**Rating:** 7
**Confidence:** 3

**Summary:**

The authors propose a pseudo-semantic loss for deep generative models with logical constraints. The authors consider autoregressive generative distributions (modeled by RNNs, Transformers, etc.), which are more expressive, and go beyond the standard approach of enforcing the constraints on fully-factorized distributions. However, the likelihood of the constraint is hard for auto-regressive distributions, so the authors propose an approximation at a random, local point, yielding a "pseudolikelihood". Then they show the utility of that pseudolikelihood in constructing a pseudo-semantic loss, which improves standard tasks such as finding the shortest path on Warcraft maps and solving partially solved 9x9 Sudokus. As a "killer app," the authors propose to use their method on detoxifying generations from LLMs. They show results on detoxifying GPT-2.



**Strengths:**

S1: The problem of adding constraints with semantic losses is important, and has a clear impact on relevant problems, such as reducing toxicity.

S2: The approximations in the paper are sound and seem to be of good fidelity and low deviation from the ground truth.

S3: The experiments are performed on a variety of tasks, showing the promise of the method.

**Weaknesses:**

W1: The authors need to give more examples of constraints, logical circuits, and how the loss encourages the reuse of sub-problems. See my questions below for more concrete suggestions for improvement. Furthermore, I do not think that Figure 2 is a good example. The choice of numbers at the leaves is arbitrary (should be discussed). The computation of the probabilities on the left is not clear. It's not clear how one relates A, B, and C to the constraint with Cat, Animal, and Dog. Some simplification and/ or explanation of the steps are necessary.

W2: The reader does not get a sense of how scalable the approach is. It is acknowledged in the paper that the circuits' complexity is a bottleneck (260-263). I would expect more discussion on the circuits used in the paper, and how they could scale. What more complicated problems should we expect to solve by using the proposed pseudo-semantic loss?

W3: Have you tried other constraints for toxicity? Was the avoiding tokens from the list the first thing you tried? Why not try more stuff to check for more gains? Again, it's important to discuss the circuits design and the issues that arise.

Minor:

M1: line 115 - do you mean $\mathbf{w}_i$ in the softmax? I do not see where the index $j$ comes from.

M2: line 9 in Algorithm 1 - I think you mean "seq" vs, "seq_len" and "cats" vs. "num_cat"?

M3: line 323: one again <- once again.

M4: line 508: generate 100k <- generate 100k samples (or generations)?

**Questions:**

Q1: Not sure I understood how your approach reuses sub-problems. Could you give examples from the tasks you use?

Q2: Could you give examples of reducing toxicity by looking at samples from the GPT model?



**Limitations:**

the authors adequately addressed the limitations

---

> ### Author Rebuttal · Authors · 2023-08-09
>
> We would like to thank the reviewer for engaging with our work and their valuable feedback.
>
> We are happy to see them acknowledge the importance of the problem tackled and the value of both the proposed approximation and experimental evaluation.
>
> [“ It's not clear how one relates A, B, and C to the constraint with Cat, Animal, and Dog.“]
>
> We agree that a mapping from the variable A, B and C to Cat, Animal and Dog is missing.
> We will modify the example to either make the mapping explicit, or only use one or the other.
> The implied mapping is that variable A maps to Cat, variable B maps to dog and variable C maps to Animal. We will make the mapping clear and/or modify the example accordingly.
>
> [“The authors need to give more examples of constraints, logical circuits”]
>
> A logical constraint is simply any sentence in Boolean logic. For instance, a very simple constraint might be $y_3, = \text{dog}$ which simply states that all sentences must have as their third token “dog”. In the example we presented, our constraint simply asserts that if I predict an “cat” or “dog”, I necessarily need to predict an animal (every “cat” is an animal, and every “dog” is an animal; If I acknowledge the existence of either in an image, logic necessitates that I also acknowledge the existence of an animal in the image.
>
> A logical circuit is  a computational graph for compactly representing solutions of  a constraint, that is all possible assignments where the constraint holds true. There every AND node represents a (partial) solution to the constraint and every OR node represents a distribution over mutually exclusive and exhaustive (partial) solutions. More concretely, focusing on the upper-right AND node, we can see that it represents the Boolean function $$(\text{Animal} \lor \lnot \text{Animal}) \land (\lnot \text{Cat} \lnot \text{Dog})$$, meaning if we predict neither a dog nor a cat, then we're free to predict or not predict an animal.
>
> [“how the loss encourages the reuse of sub-problems”]
>
> We simply mean the solution of a problem depends on already-computed solutions to other subproblems in a dynamic-programming fashion. Take for example the problem of choosing $k$ out of $n$ elements which has a simple structurethe probability of selecting $k$ out of $n$ elements is simply the probability of selecting $k-1$ out of $n-1$ elements AND selecting the current element OR selecting $k$ out of $n-1$  elements AND NOT selecting the current element.
>
> [“The choice of numbers at the leaves is arbitrary (should be discussed). The computation of the probabilities on the left is not clear”]
>
> Back to our example, the numbers on the second row on the left, denoted by an arrow "eval" are the likelihood assigned by the auto-regressive model to these joint assignments i.e. they are the output of a neural network. The row below that, denoted by an arrow "norm" refers to the conditional probabilities, obtained through normalizing the joint by the marginal e.g. $p(a|b,c) = p(a,b,c) / p(b,c)$ on the right, the only numbers we supply are those at the leaves $A, \lnot A, B, \lnot B, C$ and $\lnot C$ .These correspond exactly to the conditionals obtained on the last row on the left.
>
> [“The reader does not get a sense of how scalable the approach is. It is acknowledged in the paper that the circuits' complexity is a bottleneck (260-263). I would expect more discussion on the circuits used in the paper, and how they could scale. What more complicated problems should we expect to solve by using the proposed pseudo-semantic loss?”]
>
> The size of the logical circuit can indeed worst-case grow exponentially in the size of the constraint. That being said, there are some problems that exhibit a surprising amount of structure theoretically guaranteeing compact circuit representations, examples of which are the $k$-subset constraint and perfect matching on planar graphs. For many other problems, we can obtain compact logical circuits for many instances of interest e.g. entity-relation extraction, NLI transitivity, MNIST addition, and many many more. There are cases in practice, of course, when the circuits grow too much beyond what is computational feasible, in which case we can resort to approximating the logical circuit [1, 2] which should work seamlessly with our distributional approximation.
>
> [“What more complicated problems should we expect to solve by using the proposed pseudo-semantic loss?”]
>
> We expect pseudo-semantic loss to be of use in any task where we have domain knowledge relating the outputs of a classifier (also known as structured-prediction problems), and where we have that the output distribution induced by the classifier goes beyond the fully-factorized distribution.
>
> [“other constraints for toxicity? "]
>
> We have not explored other constraints for toxicity. It is generally not easy to capture toxicity using a logical constraint as the trait of being toxic can be attributed to latent factors beyond just the presence of certain keywords e.g. a condescending tone. Instead, our hope was that steering the model's distribution from sentences containing toxic words would serve to steer it away from toxic sentences in general as both are typically correlated. This is a direction that we hope to explore in future work.
>
> [“examples of reducing toxicity”]
>
> Please see the PDF included in the general response.
>
> [“Typos”]
>
> Thank you for pointing out the typos. We will be sure to correct them in the camera-ready.
>
> References:
>
> [1] Kareem Ahmed, Kai-Wei Chang and Guy Van den Broeck. 2023. Semantic Strengthening of Neuro-Symbolic Learning. In Proceedings of International Conference on Artificial Intelligence and Statistics (2023).
>
> [2] Robin Manhaeve, Giuseppe Marra, Luc De Raedt. 2021. Approximate Inference for Neural Probabilistic Logic Programming. In Proceedings of the 18th International Conference on Principles of Knowledge Representation and Reasoning.

---

> > ### Comment · Reviewer_P5zX · 2023-08-19
> > **Thanks for the clarifications.**
> >
> > The authors have addressed my questions and comments well. I am maintaining my positive score. Regarding the discussion with the other reviewers, there was a question of whether RNNs/ LSTMs are autoregressive. Yes, they are. I think the reviewers responded well to that.

---

### Official Review · Reviewer_FKar · 2023-07-01

**Soundness:** 3 good
**Presentation:** 3 good
**Contribution:** 3 good
**Rating:** 6
**Confidence:** 4

**Summary:**

The paper presents a new way of computing a loss function measuring the degree of satisfaction of logical constraints for autoregressive models. This is an important task, as autoregressive models are now more and more used and calculating the degree of satisfaction of even simple constraints made of a single literal is #P-hard (due to the fact that different solutions depend on different conditionals). The proposed solution has been evaluated on 3 different tasks:

1. Warcraft shortest path: the task is to generate a minimum-cost path from the upper left to the lower right vertices.

2. Sudoku: the task is to generate a solution to a given partially filled sudoku

3. LLM detoxification: the task is to steer away the LLM model from toxic prompted-generations.

**Strengths:**

- The paper is very well written and easy to follow.

- The introduced approach is very interesting and presents improvements in the experiments.

- The problem solved is very important as auto-regressive models are becoming more and more important (mostly due to the spread of LLMs)

**Weaknesses:**

- The authors claim that they are the first to learn with constraints under auto-regressive generative models. While this might be true, there has been a lot of work on steering the generation of auto-regressive LLMs at inference time (see, e.g., [1,2]). I think that discussing the pros and cons of learning to satisfy the constraints during training vs steering the generation during inference might be very beneficial for the community. Especially given the fact that the authors have a NLP task in their experiments.

- On this note, it might be interesting to see how their method performs against such methods in the LLM detoxification task.

- I think that the current title is a bit misleading. Not all deep generative models are auto-regressive. So I would change the title from "A Pseudo-Semantic Loss for DGM with Logical Constraints" to  "A Pseudo-Semantic Loss for auto-regressive models with Logical Constraints".

References:

[1] Meng et al. Controllable text-generation with neurally-decomposed oracle, NeurIPS 2022.

[2] Lu et al. NeuroLogic Decoding: (Un)supervised Neural Text generation with predicate logic constraints. ACL 2022


**Questions:**

- I found the entire paper very clear. However, I have a doubt about this step at equation 7. Why can you write:
$\mathbb{E}_{y \sim \tilde p}[\mathbb{1}(y \models \alpha)] \approx \mathbb{E}_{y\sim p} \mathbb{E}_{\tilde y \sim \tilde p_y}[\mathbb{1}(\tilde y \models \alpha)]$?
Can you give some more details about this step?

- Algorithm 1: what is the difference between seq and seq\_len? In the same way, what is the difference between cats and num\_cat? Also, expand works on non-singleton dimensions. Since you're having python code in the paper then it needs to be as you'd write it.

- In the example, suppose that instead of having generated $abc$, the models has generated the phrase "I love dogs", the samples that are 1 hamming distance away then would be "love dogs", "I dogs" and "love dogs"? Or something else?

- Again in the example, you use a constraint with Cat, Dog and Animal as variables. Can you rewrite it using $a,b,c$. Also, in appendix A, it would be nice to have the step by step guide showing how the circuit is built from the constraint.

- Minor: in the related work it shouldn't be HMCCN but C-HMCNN

P.S. I really like the paper, if all the questions can be answered well, then I will increase the score.

**Limitations:**

See weaknesses.

---

> ### Author Rebuttal · Authors · 2023-08-09
>
> We would like to thank the reviewer for their thoughtful comments.
>
> We are delighted they found the paper easy to follow, the problem to be timely, and the approach to be interesting.
>
> ["The authors claim that they are the first to learn with constraints under auto-regressive generative models. While this might be true, there has been a lot of work on steering the generation of auto-regressive LLMs at inference time (see, e.g., [1,2]). I think that discussing the pros and cons of learning to satisfy the constraints during training vs steering the generation during inference might be very beneficial for the community. Especially given the fact that the authors have a NLP task in their experiments… On this note, it might be interesting to see how their method performs against such methods in the LLM detoxification task."]
>
> There has indeed been numerous work on controllable text generation which we’re happy to acknowledge in the related works section of our camera-ready.  We would like to point out, however, that when specifying “learning with constraints”, we mean so in the specific sense of learning to maximize the training data likelihood, through cross-entropy, subject to some constraint. This can be achieved through the addition of a *regularization* or *penalty* term that *also* ensures the network's outputs satisfy the constraint. In such a setting, all the work that we are aware of only considers fully-factorized distributions. Please see the general response regarding how learning with constraints relates to constrained generation.
>
>
> [“I think that the current title is a bit misleading. Not all deep generative models are auto-regressive."]
>
> It is true that not all deep generative models are auto-regressive. Our intention was to hint at the fact that our approach is applicable to any likelihood-based model where the output distribution is not fully-factorized. However, we agree that maybe “auto-regressive” would be more befitting, especially given our experiments.
>
> [“However, I have a doubt about this step at equation 7”]
>
> This step can be understood as performing the analogous of a first-order Taylor series expansion of a discrete function with no analytical form about a point of interest, the point here being a model sample.
>
> [“In the example, suppose that instead of having generated abc, the model has generated the phrase "I love dogs", the samples that are 1 hamming distance away then would be "love dogs", "I dogs" and "love dogs"? Or something else?”]
>
> They would be all possible completions of "_ love dogs", "I _ dogs" and "I love _", where _ denotes a blank to be filled in. We can, however, save some computations by only considering the tokens of interest. More specifically, in the detoxification experiment, only the ~800 possibly-toxic tokens in addition to a single ``non-toxic'' token to fill in each blank.
>
> [“Algorithm 1: what is the difference between seq and seq\_len? In the same way, what is the difference between cats and num\_cat? Also, expand works on non-singleton dimensions. Since you're having python code in the paper then it needs to be as you'd write it.”]
>
> These are unfortunate typos that will be corrected in the camera-ready. We will map seq_len and num_cats to seq and cats, respectively.
>
> Regarding expand, you are absolutely correct, it does only operate on singleton dimensions, meaning our code would require unsqueezing the last two dimensions before calling expand, which we elected to omit for brevity's sake. But we do agree that it makes sense to present working PyTorch code, and will modify the code accordingly.
>
> [“Again in the example, you use a constraint with Cat, Dog and Animal as variables. Can you rewrite it using a,b,c”]
>
> Variable A maps to Cat, variable B maps to dog and variable C maps to Animal. We will make the mapping clear and/or modify the example accordingly.
>
> [“Also, in appendix A, it would be nice to have the step by step guide showing how the circuit is built from the constraint.”]
>
> Thank you for the suggestion. We will add it to the camera-ready.
>
> [“Minor: in the related work it shouldn't be HMCCN but C-HMCNN”]
>
> Will make sure to fix it in the camera-ready.
>
> We are happy to answer any more questions or concerns you might have.

---

> > ### Comment · Reviewer_FKar · 2023-08-13
> >
> > I would like to thank the authors for their answers.
> >
> > However, I still do not understand why they cannot compare against approaches (e.g., neurologic decoding) proposed for controllable text generation in the LLM detoxification example. Actually, I think it would be very interesting to:
> >
> > 1. compare the approaches, and
> > 2. try to see what happens when used together.
> >
> > This should be possible because the LLM detoxification example includes only the simple constraint "a list of profanity, slurs, and swear words should not appear as part of the model generations", which can also be dealt with methods such as NeuroLogic decoding.
> >
> > Also, in the general answer to all reviewers, the authors write that they can deal with non-lexical constraints. Can the authors give an example of non-lexical constraint that they can deal with in the *text generation* domain?
> >
> > Finally, will the authors change the title? If so, how?

---

> > > ### Author Response · Authors · 2023-08-14
> > >
> > > Thanks again for your response.
> > >
> > > [neurologic decoding]
> > >
> > > Here are the numbers comparing against neurologic, after painstakingly getting the code to work (the code base was quite outdated).  Results that are not significantly worse than the competition, as determined by a t-test, are boldfaced.
> > >
> > > | Method                  | Full | Toxic | Non-toxic |
> > > |-------------------------|---------------|----------------|--------------------|
> > > | GPT-2                                   | 0.11 +- 0.15       | 0.69 +- 0.13       | 0.09 +- 0.19       |
> > > | GPT-2 + neurologic              | 0.08 +- 0.14       | 0.66 +- 0.13       | **0.06 +- 0.08**       |
> > > | GPT-2 + word banning         | 0.12 +- 0.16       | 0.69 +- 0.13       | 0.09 +- 0.11       |
> > > | PseudoSL                            | **0.06 +- 0.09**  | 0.59 +- 0.04       | **0.06 +- 0.08**       |
> > > | PseudoSL + neurologic       | **0.05 +- 0.10**  | 0.68 +- 0.15       | **0.05 +- 0.07**       |
> > > | PseudoSL + word banning | **0.06 +- 0.09**  | **0.58 +- 0.01**  | **0.06 +- 0.08**       |
> > >
> > > We observe that while neurologic does improve the toxicity of the GPT-2 model, it fails when used in conjunction with our trained fine-tuned model. We do, however, show that a simple decoding method, where we set the probabilities of toxic words to 0 at decoding time, improves upon our results which is in line with our expectations given previous work.
> > >
> > > As a side note, these results are reported on a random subset of size 1k of the dataset. Attempting to run neurologic decoding on the entire dataset, 100k, using the maximum batch size we could fit on a 48GB GPU yielded an estimated time of 165 hours.
> > >
> > > [Title]
> > >
> > > We will be changing the title to "A Pseudo-Semantic Loss for Autoregressive Models with Logical Constraints".
> > >
> > > [Constraints]
> > >
> > > If we restrict ourselves to the definition in the neurologic paper "lexical constraints i.e. which words should or shouldn’t be included in the output text.", then we can specify more general constraints on the generated sentences such as "every fourth word has to be the same" or "every sentence needs to start with a noun". Essentially, any constraint that can be expressed in propositional logic can be dealt with.
> > >
> > > Thanks again for continuing to engage with us, and we hope this convinces the reviewer to push their scores towards an accept.

---

### Official Review · Reviewer_5z8g · 2023-07-06

**Soundness:** 3 good
**Presentation:** 3 good
**Contribution:** 2 fair
**Rating:** 6
**Confidence:** 3

**Summary:**

The method proposes as new neuro-symbolic loss for autoregressive models.

The proposed method approximates the ture expectation of the constraint satisfaction using a pseudo likelihood term computed only in a neighbourhood of the node.

**Strengths:**

I think the paper proposes an interesting extension of neurosymbolic methods on auto-regressive models addressing some tractability problems.

Moreover, I really appreciated the toxic language experiment and it really shows an important application  of neurosymbolic in the current AI scenario

**Weaknesses:**

My first concern/doubt is about the exploitation of RNNs as auto-regressive probabilistic model (i.e. p(y_i | y_{<i})). While RNNs have an autoregressive nature in the latent space, they do not model an auto-regressive probabilistic model.  What I mean is that, in terms of the y random variables (i.e. output variables), RNNs are fully factorised models conditioned on the parameters and the inputs, which have to be considered as observed variables in a PGM sense. Therefore it is not clear to me whether and how an LSTM is used for modelling p(y_i | y_{<i})), and whether many of the approximations are needed for an LSTM classifier. This question is also related to the baselines of the experiments, as one would be interested in knowing how pure semantic loss would behave on an RNN/LSTM.


My second doubt/question is about the fact that peudolikelihood approximations have been quite standard in statistical relational learning (e.g. MLNs/PSL inference based on pseudolikelihood) and it is not clear to me how the proposed approximations are positioned/inspired/different w.r.t. these methods.

**Questions:**

1) How is the LSTM used to model p(y_i | y_{<i})?
2) Do you have any intuition / experiment on how an RNN/LSTM behaves with pure semantic loss?
3) How is the proposed method linked to the use of pseudolikelihoods in statistical relational models (e.g. MLN learning with pseudolikehood?)

**Limitations:**

No explicit limitations are mentioned in the paper.

---

> ### Author Rebuttal · Authors · 2023-08-09
>
> We would like to thank the reviewer for the valuable feedback and for engaging with our work. We are particularly happy with their appreciation for the language detoxification experiment, which we believe has great potential for real-life impact.
>
> ["My first concern/doubt is about the exploitation of RNNs as auto-regressive probabilistic model (i.e. $p(y_i | y_{<i}))$. While RNNs have an autoregressive nature in the latent space, they do not model an auto-regressive probabilistic model. What I mean is that, in terms of the y random variables (i.e. output variables), RNNs are fully factorised models conditioned on the parameters and the inputs, which have to be considered as observed variables in a PGM sense. Therefore it is not clear to me whether and how an LSTM is used for modelling $p(y_i | y_{<i}))$, and whether many of the approximations are needed for an LSTM classifier"]
>
> It is true that the outputs $y_i$ are \emph{conditionally independent} given the hidden states $h_i$. This, however, does not imply that $p(y_1, \ldots, y_n) \neq \prod_i p(y_i | y_{<i})$.
> It also does not buy us much since we do not have oracle access to the hidden states $h_i$ ahead of time. Rather, for some weight matrices $\mathbf{U}, \mathbf{W}$ and $\mathbf{V}$ we can only obtain $\mathbf{h_i} = g(\mathbf{U} \mathbf{h_{i-1}} + \mathbf{W} \mathbf{e_i})$ and consequently $p(y_{i} | y_{t-1}) = \mathsf{Softmax}(\mathbf{V} \mathbf{h_i})$ only having processed $h_{i-1}$, and inductively, the entire prefix. Intuitively, the main computational hurdle remains: the probability $y_i$ is dependent on the entire prefix
> $y_0, \ldots, y_{i-1}$. Computing the probability of a Boolean constraint constituting of even a single term asserting e.g. $y_i$ = dog, which reduces to computing the \emph{marginal} probability $p(y_i=\text{dog})$ requires summing over all $50256^{i-1}$ prefixes ($50256$ being the number of possible tokens at every time step) , a summation which is highly intractable even for a prefix of size $10$.
>
> ["How is the LSTM used to model $p(y_i | y_{<i})$?"]
>
> Consequently, the RNN/LSTM is trained in precisely the same manner as you would train an autoregressive model i.e. we simply train the model to minimize the error in predicting the true next word in the training sequence given the prefix, using cross-entropy as the loss function (see e.g. https://web.stanford.edu/~jurafsky/slp3/9.pdf, section 9.2 for a treatment of RNNs as language models). The only modification is that we also condition on the input (Warcraft map image and the input Sudoku puzzle in the shortest path and Sudoku tasks, respectively) in a fashion very similar to CNN-BiLSTM models, throughout the entire sequence.
>
> ["Do you have any intuition / experiment on how an RNN/LSTM behaves with pure semantic loss?"]
>
> One intuition to our approach is that it is the analog of a first-order approximation to discrete functions with no analytical form. As such, our approach makes use of first-order information to approximate the value of the discrete function about a given point, in our case the model sample. One naïve approach to employing semantic loss would be to assume the distribution is a fully-factorized one and use the the conditional probabilities of the model sample $p(y_i | y_{<i})$ as marginal probabilities $p(y_i)$. It is unclear, however, what the semantics of the loss would be in this case, or in what way the number it yields relates to the original quantity. Nonetheless, on attempting to train with semantic loss, we achieved an exact match of $63$% and a consistency of $75$%, the former metric being slightly better than the baseline and the latter being slightly worse (see numbers in paper)
>
> [How is the proposed method linked to the use of pseudolikelihoods in statistical relational models]
>
> Our approach was inspired by such methods, although the end goal is at odds in a sense.
> Precisely, in statistical relational learning, maximizing the pseudolikelihood in place of the likelihood sidesteps the computation of the, very often intractable, partition function.
> In our case, however, the partition function *is* precisely what we're interested in computing, or at the very least estimating. Towards that end, the role of pseudolikelihood is as a stepping stone to help “massage” the autoregressive distribution into a form that is amenable to the reuse of computations, in a fashion similar to dynamic programming. To see what we mean by reuse of computations, consider e.g. in the easiest case, when we have a fully-factorized distribution, the probability of every sentence where $y_i$ is true depends on the same probability $p(y_i)$.
>
> [“No explicit limitations are mentioned in the paper.”]
>
> Please see section F of the appendix for some potential limitations of our proposed approach.

---

> ### Author Response · Authors · 2023-08-16
>
> We wanted to ensure we've satisfactorily addressed the reviewers concerns.
>
> To that end, we wanted to emphasize that, just like the output distribution defined by GPT, the output distribution defined by an RNN is auto-regressive. Simply put, to obtain the hidden state at time $t$, the RNN needs to have processed the entire prefix up to time $t-1$. $h_t$, and consequently $y_t$,  is then a function of the current token $x_t$ and the prefix through $h_{t-1}$. This is very similar to the
> way in which, in GPT, $y_t$ is a function of the current token and the prefix through the attention at time $t$.

---

> > ### Comment · Reviewer_5z8g · 2023-08-21
> >
> > I think this depends on how the RNN is designed, and this was my main question.
> >
> > If the output at time (t-1) is fed as input to timestep (t), then, yes, I agree that there exist a conditional dependence of the variable y_t on y_{t-1}. This is indeed what you would do for text generation. But, this is not a general property of recurrent networks. As for general definition, when the x and y space are disjoint, the only dependence of the output is on the state h_{t-1} and the history input x_{0:t}, not on the previous output y, which would make it autoregressive on this space.
> >
> > Reading the answer of the authors it seems this is the way it is implemented. I missed this detail in the paper and I think it is fundamental to give credit to the experimental campaign.
> >
> > I will therefore change my score accordingly.

---

### Official Review · Reviewer_sCK6 · 2023-07-27

**Soundness:** 1 poor
**Presentation:** 1 poor
**Contribution:** 1 poor
**Rating:** 3
**Confidence:** 4

**Summary:**

This paper proposes adding a pseudo-semantic loss into the training of autoregressive models, so that the model can learn logical constraints. Specifically, this approach includes a data augmentation by perturbation (e.g., by Hamming distance), and then adding a penalty loss for generation steps that are not following given constraints.

**Strengths:**

I don't see obvious strengths.

**Weaknesses:**

* The writing of this paper is of low quality. For example, the meaning of Figure 1 is vague. As a front-page figure, it is even harder to understand than the  algorithm pseudo-code.
* The most important claim of this paper, "the first approach to learning with constraints under auto-regressive generative models", is actually not true. Instead, there is already a bunch of previous work on the topic of "generation with constraints", such as [1] and its followup work.

[1] NeuroLogic Decoding: (Un)supervised Neural Text Generation with Predicate Logic Constraints (NAACL 2021)

**Questions:**

I don't have a question in this stage.

**Limitations:**

I don't see the discussion of limitations in this paper.

---

> ### Author Rebuttal · Authors · 2023-08-09
>
> Thank you for your response.
>
> [The writing of this paper is of low quality. For example, the meaning of Figure 1 is vague. As a front-page figure, it is even harder to understand than the algorithm pseudo-code.]
>
> The figure is just meant to convey our approach in broad strokes. An autoregressive model defines a distribution over the output space, shown in black. It places probability mass on sentences that both violate (white area under curve) as well as those that satisfy the constraint (pink area under curve). Our aim throughout the paper is to shift the distribution, so the autoregressive model places all its mass only on sentences that satisfy the constraint. Typically, we can achieve this by simply maximizing the probability of the constraint w.r.t the output distribution. The problem is computing the probability of the constraint w.r.t the output distribution is intractable when the distribution is autoregressive. What we propose is a local, tractable approximation of the original distribution (shown in grey) under which we can approximate the probability of the constraint. It is local in the sense of being correct within a neighborhood of a sample (the red cross). This is similar to how we compute a linear approximation of a function about a point, except here we have a discrete function with no analytical form.
>
> [The most important claim of this paper, "the first approach to learning with constraints under auto-regressive generative models", is actually not true. Instead, there is already a bunch of previous work on the topic of "generation with constraints", such as [1] and its followup work.]
>
> There has indeed been numerous work on controllable text generation which we’re happy to acknowledge in the related works section of our camera-ready.  We would like to point out, however, that when specifying “learning with constraints”, we mean so in the specific sense of learning to maximize the training data likelihood, through cross-entropy, subject to some constraint. This can be achieved through the addition of a *regularization* or *penalty* term that *also* ensures the network's outputs satisfy the constraint. In such a setting, all the work that we are aware of only considers fully-factorized distributions. Please see the general response regarding how learning with constraints relates to constrained generation.
>
> [“I don't see the discussion of limitations in this paper.”]
>
> Please see section F of the appendix for some potential limitations of our proposed approach.
>
> We are happy to engage with the reviewer regarding any misunderstandings regarding
> the paper's contribution as well as the presentation. Our hope is that this paper is accessible to a wide audience

---

> ### Author Response · Authors · 2023-08-16
>
> We wanted to re-emphasize the key difference between our work and "generation with constraints" works such as NeuroLogic. Our proposed approach is a **training time** approach that biases the function learnt from the data to respect certain constraints by adding a regularization term to the cross-entropy loss. It **does not** interfere with the generation process at test time. "generation with constraints" approaches on the other hand **do not interfere with the training process**, and only modify an **already trained** model's predictions at test time.

---

> > ### Comment · Reviewer_sCK6 · 2023-08-20
> > **Re: Official Comment by Authors**
> >
> > Thanks for your clarification on the difference between your work and NeuroLogic. However, I would like to insist my score for now, because I am still not convinced regarding the validation of the big claim "the first approach to learning with constraints under auto-regressive generative models":
> >
> > * "Learning" is a rather high-level concept, and people don't always understand it as "training", especially when it doesn't appear together with "inference", so it might be better to claim it as "training generative models with constraints".
> > * Even considering training-time approaches only, there are still a bunch of work closely related, especially [1] and its following work. After a quick reading of [1], I even feel the proposed approach in this paper can be seen as a special case of [1]. Specifically, [1] adapts a posterior regularization formulation to text generation. It has a search space approximation and a regularization term in its objective function, corresponding to the purturbation of y and the penalty loss term in this paper, respectively. There are more related work mentioned in [2], such as [3].
> > * Missing discussion and citation for all the above closely related work, including NeuroLogic, raises an overall concern on the thoroughness of the background introduction in this paper.
> >
> >
> > [1] Prior Knowledge Integration for Neural Machine Translation using Posterior Regularization (ACL 2017) \
> > [2] A Survey of Knowledge-Enhanced Text Generation (ACM Computing Survey, 2022) \
> > [3] Deep Generative Models with Learnable Knowledge Constraints (NeurIPS 2018)

---

> > > ### Author Response · Authors · 2023-08-21
> > >
> > > Thank you for your response and help towards improving the paper.
> > >
> > > ["might be better to claim it as "training generative models with constraints""]
> > >
> > > Thank you for your suggestion. We will make sure to clarify it in the paper.
> > >
> > > ["considering training-time approaches only, there are still a bunch of work closely related…, I even feel the proposed approach in this paper can be seen as a special case of [1]. "]
> > >
> > > Thank you for pointing out this related work. We are happy to discuss it in our related works section. It is unclear, however, how applicable it is to our setting. The approach in [1] rests upon the framework of posterior regularization [4], which requires that we design feature functions encoding the constraint. Such feature functions need to *factorize* over the tokens of the sentence (see section 2.5 in [4]). That might be doable for very simple constraints such as those considered in [1], but not in general.
> > >
> > > Setting aside that issue, the approach in [1] minimizes an extra KL-divergence between the autoregressive distribution and a log-linear variational distribution. For the constraints considered, computing such KL-divergence is intractable. Even approximating the KL-divergence is hard, due to the computational hardness of sampling the constraint ([1] sample from the autoregressive distribution and threshold using the feature function, which does not sample from the correct distribution). There is also no notion of perturbations; these are simply samples from the distribution. We would also argue that log-linear models defined by a set of linear constraints as variational distributions have been in use since posterior regularization was first introduced. [1] is simply an instantiation of such a framework. In our case, however, our distributional approximation is novel: it is a first-order approximation of the true autoregressive distribution. Consequently, we are **exactly computing the probability of the constraint being satisfied w.r.t. our proposed approximate distribution**.
> > >
> > > ["Missing discussion and citation"]
> > >
> > > We are happy to cite any and all related work. We were of course aware of neurologic and other inference time approaches, but felt they might be orthogonal to approach proposed here. Having discussed it with the reviewers, however, we will happily include a discussion with such works. Moreover, our work tends to focus on developing methodology for *general* constraints. And despite our best efforts, due to the massive number of works, it can be easy to miss methods geared towards very specific tasks from other communities. We greatly appreciate the reviewer bringing these to our attention.
> > >
> > > ["Claims"]
> > >
> > > Having said all of the above, fundamentally, the issue at hand seems to be the reviewer's disagreement with our overarching claim. We are happy to relax our claim to something along the lines of "Unlike previous works that are only able to approximately handle simple constraints under relaxations of autoregressive distributions, our approach injects non-trivial constraints, that don't easily factorize, as part of the training process, computing the probability of the constraint exactly w.r.t an approximate distribution".
> > >
> > >
> > >
> > > Thanks again for engaging with us, and we hope this convinces the reviewer to push their scores towards an accept.
> > >
> > > [References]:
> > >
> > > [4] Posterior Regularization for Structured Latent Variable Models (JMLR 2010)

---

### Author Rebuttal · Authors · 2023-08-09

We would like to thank all the reviewers for their valuable feedback towards improving our paper.

We are happy to see the reviewers' excitement regarding the posed problem, the proposed solution as well as the empirical evaluation, with an emphasis on the LLM detoxification experiment.

There have been several comments regarding controllable language generation and its relevance to our approach. There has indeed been a lot of work on controllable text generation which we’re happy to acknowledge and discuss in our camera-ready. We would like to point out, however, that both approaches are not at odds. Indeed, they have been shown to be *complementary* by previous work, both old [1] and new [2] [3], theoretically and empirically, with methods combining both paradigms attaining the highest performance. In the specific context of language detoxification, [4] has shown complementing domain-adaptive training, such as our approach, with decoding-time constraints to yield the best results.
It is, however, non-trivial to generalize such decoding methods beyond *non-lexical constraints* e.g. Sudoku or Warcraft, which seem to be the focus of all such approaches.

Reviewers have also suggested adding a step-by-step construction of the logical circuit in our examples, which we are happy to add in the camera-ready.

We will now address each reviewer's individual concerns.

References

[1] Ming-Wei Chang, Lev Ratinov, and Dan Roth. 2007. Guiding Semi-Supervision with Constraint-Driven Learning. In Proceedings of the 45th Annual Meeting of the Association of Computational Linguistics.

[2] Kareem Ahmed, , Eric Wang, Guy Van den Broeck and Kai-Wei Chang. 2021. Leveraging Unlabeled Data for Entity-Relation Extraction through Probabilistic Constraint Satisfaction. ArXiv abs/2103.11062.

[3] Kaifu Wang, Hangfeng He, Tin D. Nguyen, Piyush Kumar and Dan Roth. 2023. On Regularization and Inference with Label Constraints. In Proceedings of the 40th International Conference on Machine Learning.

[4] Boxin Wang, Wei Ping, Chaowei Xiao, Peng Xu, Mostofa Patwary, Mohammad Shoeybi, Bo Li, Anima Anandkumar and Bryan Catanzaro. 2022. Exploring the Limits of Domain-Adaptive Training for Detoxifying Large-Scale Language Models. ArXiv abs/2202.04173.

---

> ### Author Response · Authors · 2023-08-15
> **Update: Added comparison to neurologic decoding and word banning**
>
> Here are the numbers comparing against neurologic and word banning. Results that are not significantly worse than the competition, as determined by a t-test, are boldfaced.
>
> | Method                  | Full | Toxic | Non-toxic |
> |-------------------------|---------------|----------------|--------------------|
> | GPT-2                                   | 0.11 +- 0.15       | 0.69 +- 0.13       | 0.09 +- 0.19       |
> | GPT-2 + neurologic              | 0.08 +- 0.14       | 0.66 +- 0.13       | **0.06 +- 0.08**       |
> | GPT-2 + word banning         | 0.12 +- 0.16       | 0.69 +- 0.13       | 0.09 +- 0.11       |
> | PseudoSL                            | **0.06 +- 0.09**  | 0.59 +- 0.04       | **0.06 +- 0.08**       |
> | PseudoSL + neurologic       | **0.05 +- 0.10**  | 0.68 +- 0.15       | **0.05 +- 0.07**       |
> | PseudoSL + word banning | **0.06 +- 0.09**  | **0.58 +- 0.01**  | **0.06 +- 0.08**       |
>
> We observe that while neurologic does improve the toxicity of the GPT-2 model, it fails when used in conjunction with our trained fine-tuned model. We do, however, show that a simple decoding method, where we set the probabilities of toxic words to 0 at decoding time (through HuggingFace), improves upon our results which is in line with our expectations given previous work.
>
> As a side note, these results are reported on a random subset of size 1k of the dataset. Attempting to run neurologic decoding on the entire dataset, 100k, using the maximum batch size we could fit on a 48GB GPU yielded an estimated time of 165 hours.

---

### Decision · Program_Chairs · 2023-09-21

**Decision:**

Accept (poster)

**Comment:**

This paper proposes a new approach for training auto-regressive models (e.g., language models) with logical constraints. A new neuro-symbolic loss is developed with various approximations to make the loss tractable and efficient for training. Evaluated on three tasks (Warcraft shortest path, Sudoku, and LM detoxification), the approach shows improvements. The reviewers acknowledged the problem of adding constraints into auto-regressive model training is important, and the proposed loss and approximations are sound.

A major concern is that the paper claimed itself to be “*the first approach to learning with constraints under auto-regressive generative models*”, which is probably not true as pointed out by the reviewers. There are previous general approaches of learning with constraints, such as posterior regularization [1-4] (e.g., [4] applied non-factorizable constraints on training auto-regressive text generation models). Additionally, as again pointed out by the reviewers, there are also a bunch of works of inference-time control of language model generations, such as [5, 6], where [5] studies specific logical constraints and [6] studies general forms of constraints, both applying to auto-regressive LMs. Though these works focus on adding constraints during inference instead of training, they are very relevant to the submission and worth thorough discussion.



[1] Posterior Regularization for Structured Latent Variable Models (JMLR 2010)

[2] Prior Knowledge Integration for Neural Machine Translation using Posterior Regularization (ACL 2017)

[3] A Survey of Knowledge-Enhanced Text Generation (ACM Computing Survey, 2022)

[4] Deep Generative Models with Learnable Knowledge Constraints (NeurIPS 2018)

[5] NeuroLogic Decoding: (Un)supervised Neural Text Generation with Predicate Logic Constraints (NAACL 2021)

[6] COLD Decoding: Energy-based Constrained Text Generation with Langevin Dynamics